# CO-COME: A Contrastive Label Disambiguation Framework with Combined MTS Feature Encoder for Partial-Label Multivariate Time Series Classification

## Abstract

Multivariate Time Series Classification (MTSC) is a conventional time series task applied in the fields of finance, healthcare, and weather forecasting. However, it is often plagued by a lack of high-quality labels in practical applications. To address the label quality issues in real-world scenarios, the Partial Label Learning (PLL) paradigm has been proposed. This paradigm solves the problem of ambiguous labels by allowing each training instance to be associated with a set of candidate labels. The superiority of PLL has been verified in the field of image classification. But due to the inherent difficulties in feature extraction for Multivariate Time Series (MTS) and the lack of appropriate data augmentation strategies, PLL has not been applied in MTSC tasks. Motivated by this, we propose a novel model: **CO**ntrastive label disambiguation framework with **CO**mbined **MTS** feature **E**ncoder (**CO-COME**), which integrates a contrastive learning-based label disambiguation framework with an efficient MTS feature representation encoder, CTFE. The contrastive learning module leverages label prototypes to effectively resolve label ambiguity under the PLL setting, meanwhile the CTFE encoder is designed to capture both explicit and latent representations of time series data, enabling robust and discriminative feature learning. Extensive experiments on 20 UEA benchmark datasets demonstrate that our model achieves state-of-the-art performance under partial-label conditions. Our method will be available in `https://github.com/Noname9971/CO-COME` once accepted.

## 1 Introduction

MTS is an important type of data that covers a wide range of areas such as disease diagnosis, traffic analysis, financial prediction and so on. MTSC is one of the most fundamental tasks of MTS. It is challenging due to diverse temporal dependencies and high dimensionality, which constrain performance and deployment in real-world settings. MTSC has developed rapidly in recent years. Traditional methods such as the distance-based methods: Dynamic Time Warping (DTW) with 1-Nearest Neighbor (1-NN) Bagnall et al. (2018) and feature-based methods: the bag of Symbolic Fourier Approximation (SFA) Schäfer & Högqvist (2012) have shown good performance in the MTSC tasks. Traditional MTSC methods rely on manual preprocessing and handcrafted features. They work on small-scale or domain-specific tasks, but these predefined features generalize poorly and are unsuitable for most datasets. MTS data usually contains more implicit features which are often difficult to manually capture and assess. The development of deep learning methods solve this problem to a certain extent. Convolutional neural networks (CNNs) are generally adopted for extracting local features. LSTM Hochreiter & Schmidhuber (1997), GRU Chung et al. (2014), and Transformer Vaswani et al. (2017) can automatically recognize the features from time domain information and analyze the latent relation between them. Specific networks derived from these basic networks, such as TapNet Zhang et al. (2020) and Densely Knowledge-Aware Network Xiao et al. (2024), have achieved significant improvements.

Deep neural networks often require large amounts of labeled data. However, collecting accurate labels is challenging, especially in real-world applications. Labels are usually ambiguous or noisy.

This issue is particularly serious in time series data, which lack visually intuitive patterns for humans to recognize. As a result, labeling often depends on domain experts and is prone to errors. Label ambiguity is common in MTSC, yet it is often overlooked by existing research. Consequently, many state-of-the-art MTSC methods that achieve best on academic datasets don't perform well on industrial data due to their inability to handle ambiguous labels.

To address label ambiguity in MTSC, we adopt partial-label learning (PLL) Zhang et al. (2017), where each training instance is associated with a candidate label set that typically contains the true class. Although PLL is well studied in vision tasks, it remains underexplored for MTSC due to challenges in representation learning and augmentation for multivariate sequences. We propose CO-COME, which integrates PLL with a contrastive label-disambiguation framework and a combined time series feature encoder (CTFE), enabling robust classification under ambiguous labels. To effectively address the label ambiguity inherent in PLL, we draw inspiration from MOCO He et al. (2020) and PiCO Wang et al. (2022) by introducing a contrastive learning framework coupled with label prototypes to perform label disambiguation. Additionally, we design a dynamic multivariate time series data augmentation module (DMDA), which adaptively adjusts augmentation intensity according to dataset. To further enhance representation learning, we introduce CTFE, an encoder that integrates inherent and explicit features with trainable latent representations for robust time series encoding. In summary, the main contributions of this work are as follows:

- **First work that applies PLL to MTSC**: To our knowledge, this is the first work that effectively applies PLL to MTSC, explicitly training with candidate-label sets under label ambiguity. Our method achieves state-of-the-art performance under the PLL condition.

- **A contrastive label disambiguation framework with DMDA**: We introduce a PLL framework for MTSC that couples a contrastive learning strategy. Dynamic Multivariate Data Augmentation (DMDA) estimates dataset-level inter-channel correlations and adjusts augmentation strength accordingly, then produces two complementary views per sample: a weak augmented view (W) and a strong augmented view (S) via calibrated operations. Using these S–W pairs in a contrastive objective, the framework learns robust, ambiguity-tolerant representations under PLL conditions.

- **A comprehensive feature encoder CTFE**: CTFE combines inherent, explicit features and trainable, latent features as the MTS feature representation. The combination of explicit features and trainable latent features effectively enhances the overall performance of the framework.

## 2 RELATED WORK

### 2.1 LABEL AMBIGUITY AND PARTIAL LABEL LEARNING

Label ambiguity refers to the situation that a training sample is associated with uncertain, noisy, or inconsistent labels. This situation is very common in real-world applications where labels are collected through weak supervision, crowdsourcing, or other poor quality labeling ways. Tarekegn et al. (2024) Noise-model-based and noise-model-free methods have been proposed to learn from noisy labels. Noise-model-based methods attempt to model the latent noise distribution of labels, Yan & Guo (2021) leveraging it to reduce the adverse effects caused by label ambiguity. The noise-model-free methods try to develop inherently noise-robust strategies using a robust loss function or a regularizer. Yu et al. (2018); Yan & Guo (2021)

Going a step further, when each training sample is annotated with a set of candidate labels, among which only one is guaranteed to be the ground-truth, this condition is called Partial Label Learning (PLL). Generally, the PLL methods can be divided into two categories: one is conventional feature engineering with simple classifier, and the other is end-to-end deep learning method. Liu et al. (2024)

The conventional methods inculde two frameworks: the average-based and identification-based disambiguation framework. The average-based disambiguation strategy treats each candidate label equally during model learning. Zhang & Yu (2015) The identification based disambiguation strategy considers the ground-truth label as a latent variable. It identifies the ground-truth label by deriving confidence scores for all candidate labels. Tang & Zhang (2017); Feng & An (2019) But the fa-

tal weakness of these methods is that they heavily rely on the pre-acquired feature representations, which means their scalability to large-scale datasets is greatly limited.

To some extent, end-to-end deep learning methods can solve the reliance problem on pre-acquired feature representations thus demonstrating promising performance. For instance, Yao et al. (2020) utilizes the temporally assembled predictions on different epochs as additional supervision information to guide the training of next epoch. Based on the contrastive learning, Xia et al. (2022) learns progressively contrastive representation space based on the ambiguity-induced positive selection to extract potential information from label ambiguity. Furthermore, Wen et al. (2021) proposed a family of loss functions named leveraged weighted loss with risk consistency that considered the trade-off between losses on partial labels and non-partial ones. Some recent works have begun to address label noise in time-series settings: Zhou et al. (2024) and Atkinson & Metsis (2020) focus on univariate time-series classification with noisy labels, Nagaraj et al. (2024) studies temporally evolving noisy labels. However, these methods cannot be directly applied to multivariate time-series classification, especially under the PLL setting considered in this paper.

Nevertheless, most existing studies on PLL have primarily focused on imaging, while overlooking the important domain of time series classification. Moreover, many feature engineering techniques and end-to-end models developed for image data are not directly applicable to MTS data due to their distinct temporal and structural characteristics.

### 2.2 MULTIVARIATE TIME SERIES CLASSIFICATION

Multivariate time series classification has been extensively studied in the literature. In general, these methods can be grouped into three categories: distance-based, feature-based, and end-to-end deep learning-based methods.

Distance-based methods typically integrate Nearest Neighbor (NN) Peterson (2009); Martínez et al. (2019) and Dynamic Time Warping (DTW) Berndt & Clifford (1994); Senin (2008) to compute similarities between the spatial features of data and perform classification. A large number of DTW-NN-based algorithms have been developed for MTSC, Iwana et al. (2020) e.g., hierarchical vote collective of transformation-based ensembles (HIVE-COTE), Lines et al. (2016); Middlehurst et al. (2021); Lines et al. (2018) random interval spectral ensemble (RISE), Flynn et al. (2019) explainableby-design ensemble method (XEM). Fauvel et al. (2022) For feature-based methods, ROCKET and MiniROCKET Dempster et al. (2021) applies thousands of randomly parameterized 1-D convolutional kernels, computes simple per-kernel statistics and train a linear model for classification. Studyies indicate that both distance-based and feature-based methods provide effective evaluation for small datasets. Nevertheless, when applied to large-scale or high-dimensional datasets, these methods often struggle to extract enough features, leading to poor performance.

End-to-end deep learning-based methods encompass a variety of network architectures built upon LSTM Hochreiter & Schmidhuber (1997), CNN, and Transformer Vaswani et al. (2017) models. They have the ability to model hierarchical internal data representations by capturing the inherent relationships among features. For example, the MLSTM-FCNs Karim et al. (2019) employs an LSTM layer and a stacked CNN layer to extract features for classification. Similarly, TapNet Zhang et al. (2020) aggregates LSTM, stacked CNN, and attention prototype network to learn the latent features. OS-CNN Tang et al. (2020) designs an Omni-Scale block to cover the best receptive field size across different datasets. Recently, several stronger MTSC models have further improved representation learning by exploiting shapelet-style transformers, time-aware multiple instance learning, graph-aware contrastive objectives, and hybrid local–global architectures Le et al. (2024); Chen et al. (2024); Wang et al. (2024); Mu et al. (2025).

Nevertheless, feature representations derived from a single model are often not sufficiently robust. Deep learning methods are highly sensitive to the quality and quantity of the training data. When the dataset is small or noisy, these methods tend to see a clear decrease in performance.

## 3 METHODS

In our study, we propose CO-COME, a contrastive partial-label method for MTSC that resists label noise and delivers state-of-the-art results under PLL conditions. In this section, we will first formu-

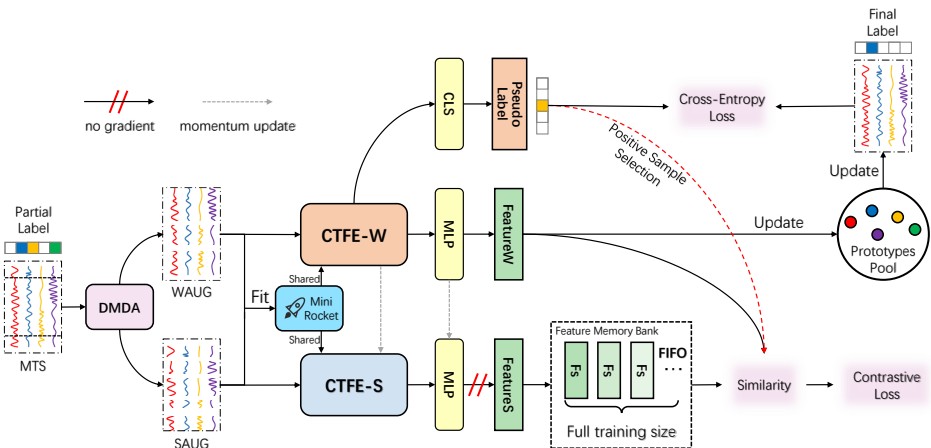

Figure 1: The overall architecture of our CO-COME model: The DMDA module applies adaptive augmentations to MTS with candidate label sets, generating weakly and strongly augmented pairs (WAUG and SAUG). WAUG and SAUG are jointly used to fit a frozen MiniROCKET. Two parallel encoders, CTFE-W and CTFE-S, initialized identically and sharing the same MiniROCKET, extract MTS representations. CTFE-S is updated via momentum from CTFE-W. The resulting features are used in a PICO-style contrastive learning framework for label disambiguation.

late the definitions of MTSC with ambiguous labels, then we will introduce the architecture of our CO-COME model.

## 3.1 PROBLEM FORMULATION

A multivariate time series $X \in R^{T \times M}$ is a sequence of real-valued vectors, $X = \{x_1, x_2, ..., x_T\}$, where $M$ is the feature dimension ,$T$ is the length of the time series. Given a collection of MTS $\mathcal{X} = \{X_1, X_2, ..., X_N\}$ of $N$ instances and its corresponding label set $\gamma = \{y_1, y_2, ...y_N\}, y_i \in \mathcal{Y}, \mathcal{Y} = \{1, 2, 3, ...C\}$,the goal of multivariate time series classification is to learn model parameters $\theta$ such that the predicted label set $\hat{\gamma} = \{f_\theta(X_i)\}_{i=1}^N$ minimizes the discrepancy to the true label set $\gamma = \{y_i\}_{i=1}^N$.

$$\theta^* = \arg \min_\theta \frac{1}{N} \sum_{i=1}^N \ell\left(f_\theta(X_i), \ y_i\right)$$

Transfering existing time series classification methods to industrial datasets would decrease the model performance due to ambiguous labels. The definition of ambiguous labels is that for each $X_i$, the previously given $y_i, y_i \in \mathcal{Y}$ may not be the actual correct label $y_i'$ . To address this issue, we refer to the idea of partial label learning: we assume a candidate label set: $Y_i \subset \mathcal{Y}$ that $y_i' \in Y_i$. Our goal is to train a functional mapping $f$ and its parameters $\theta$ on the partial label set that predicts the one true label associated with the input time series $\mathcal{X}$.

$$f_\theta : X_i \rightarrow y_i'$$

## 3.2 OVERALL ARCHITECTURE

Our CO-COME model is designed to effectively encode the MTS into feature representations and disambiguate the partial label set. The overall architecture of our proposed CO-COME model is shown in Figure 1. We adopt a partial label disambiguating framework including contrastive learning, pseudo-labeling and label prototype inspired by PiCOWang et al. (2022), MoCoHe et al. (2020), and SupConKhosla et al. (2020).

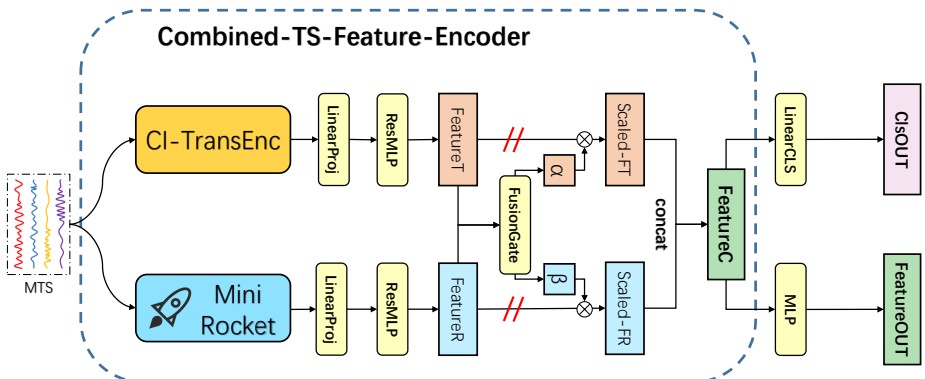

Figure 2: CTFE: For each augmentation view (WAUG, SAUG), a frozen MiniROCKET extracts PPV features $F_R$. A channel-independent Transformer encodes each channel separately to $F_T$.FusionGate computes cross-attention between $\mathbf{F}_R$ and $\mathbf{F}_T$, applies element-wise rescaling to each, and concatenates them to form $\mathbf{F}_C$.

Several specific modifications and improvements are introduced to better capture the characteristics of MTSC. First, Dynamic MTS Data Augmentation (DMDA) module (Appendix 1) will evaluate correlations of the input MTS data, set the suitable strength of data augmentation according to the correlations and generate weak and strong augmentation samples. The weak and strong augmentation samples are then respectively fed into the contrastive learning framework with two isomorphic Combined Timeseries Feature Encoder (CTFE): CTFE-W and CTFE-S. The two CTFEs convert MTS data into feature embeddings of same dimension. The Feature Memory Bank will store the $FeatureS$ of the entire dataset in a first-in-first-out order. Finally, we update the label prototypes based on the $F_W$,$F_S$, and then calculate the $L_{cls}$,$L_{cont}$

### 3.3 CONTRASTIVE LEARNING FRAMEWORK

We adopt a prototype-enhanced contrastive learning framework based on PiCO, tailored for MTSC. During training, we assign each MTS sample $x_i$ a normalized vector $s_i \in [0,1]^C$ as the pseudo target, whose entries denote the probability of labels being the ground-truth. The framework consists of a query encoder (CTFE-W) and a momentum-updated key encoder (CTFE-S) which share the same architecture but operate on differently augmented views ($WAUG$ and $SAUG$) of the input MTS data. The encoder is a customized MTS encoder that combines inherent, explicit features and trainable, latent features.

$$\theta_{CTFE-S} \longleftarrow m \cdot \theta_{CTFE-S} + (1-m) \cdot \theta_{CTFE-W}$$

Given an input MTS sample with its weak and strong augmentation view, CTFE-W produces prediction logits $z$ and a feature embedding: $FeatureW(F_w)$. To disambiguate partial labels, a masked softmax is applied using the partial label, and the pseudo label $\hat{y}$ is assigned by:

$$p = softmax(z) \odot Y, \hat{y} = \arg\max p$$

Meanwhile, CTFE-S produces a feature embedding denoted as $FeatureS(F_s)$, which is stored in a first-in-first-out (FIFO) Feature Memory Bank (FMB). FMB maintains $F_s$ for the entire training set and is used in subsequent contrastive learning. Specifically,$F_w$and $F_s$ are jointly compared with the stored features in FMB to compute the contrastive loss.

A class-specific prototype vector $\mu$ is maintained for each class $c$ and updated online using the pseudo-labeled $F_w$:

$$\mu_c = \text{Normalize}(\gamma\mu_c + (1-\gamma)F_w)$$
$$if \quad c = \arg\max f^{CTFE-W}(WAUG)$$

Moreover, the pseudo target $s_i$ is updated based on the similarity between the current feature embedding $F_s$ and the class-wise prototypes in the feature space.

$$s = \phi s + (1 - \phi)z, z_c = \begin{cases} 1, & \text{if } c = \text{argmax}_{i \in Y} F_s^\top \mu_i \\ 0, & \text{otherwise} \end{cases}$$

### 3.4 Combined Timeseries Feature Encoder

CTFE combines inherent, explicit features and trainable, latent features as the MTS feature representation. A CTFE module consists of a lightweight MiniROCKET and a channel-independent transformer encoderLiu et al. (2023b).

The channel independent (CI) transformer encoder processes each channel independently using separate normalization and transformer encoders, thus better preserving the unique temporal dynamics of each channel while reducing the risk of interference between unrelated variables. Since we apply both weak and strong data augmentations to the MTS, these operations may potentially distort the intrinsic interchannel dependencies. In contrast to cross-channel transformer architectures, the CI transformer encoder processes each channel separately, which helps to reduce the adverse impact of augmentation-induced perturbations and improves the robustness of model to channel-wise variations.

The MiniROCKET module employs a set of 84 deterministically designed kernels to compute a single statistical feature PPV. It is a stable feature extractor that does not require iterative training. To ensure feature consistency between CTFE-W and CTFE-S , CTFE-W and CTFE-S share a frozen MiniROCKET encoder with same parameters. This design enforces the extraction of common feature representations and prevents large discrepancies between the two branches, which could otherwise compromise the effectiveness of the contrastive learning framework.

The feature representation FeatureR, extracted by the MiniROCKET module, and the output FeatureC from the CI Transformer encoder are first passed through MLPs to obtain feature vectors of the same dimension. Then, fusiongate module is applied to adaptively weight and concat the two representations, resulting in the final feature FeatureOUT.

The FusionGate adapts cross-attention strategy. It applies symmetric query–key interactions between two feature vectors, generating element-wise sigmoid gates that adaptively rescale each input. This design enables fine-grained cross-modulation and enhances robustness in multivariate time series representation.

**Query projection:**

$$q_A = W_q^A A \in \mathbb{R}^d, \quad q_B = W_q^B B \in \mathbb{R}^d$$

**Key generation(per feature):**

$$K_f^B = B_f \cdot k_f^B \in \mathbb{R}^d, \quad K_f^A = A_f \cdot k_f^A \in \mathbb{R}^d, \quad f = 1, \ldots, F$$

**Scoring and gating:**

$$s_f^B = \frac{1}{\sqrt{d}} \langle K_f^B, q_A \rangle, \quad s_f^A = \frac{1}{\sqrt{d}} \langle K_f^A, q_B \rangle$$

$$g^B = \sigma(s^B) \in (0,1)^F, \quad g^A = \sigma(s^A) \in (0,1)^F$$

**Scaled outputs:**

$$A' = A \odot g^A, \quad B' = B \odot g^B$$

In CTFE-W, FeatureC is additionally passed through a classification layer to generate a pseudo label.

## 4 Results

### 4.1 Experimental Settings

**Datasets**: We utilize the UEABagnall et al. (2018) benchmark to evaluate the performance of our CO-COME model. These datasets vary in length, dimensions, and the size. Specifically, we remove

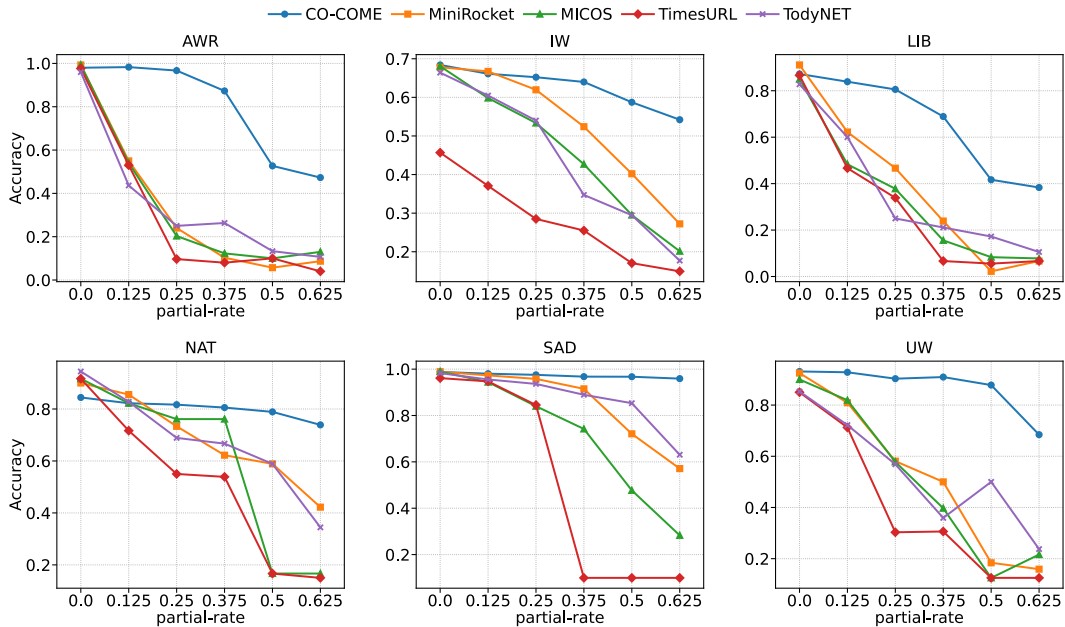

Figure 3: Accuracy on six dataset (AWR, IW, LIB, NAT, SAD, UW) under PLL condition (partial-rate=$\{0, 0.125, 0.25, 0.375, 0.5, 0.625\}$). CO-COME maintains the highest accuracy and stability compared with other models as partial-rate increases.

datasets with excessively high dimensionality or insufficient training data. Finally, we select 20 MTSC datasets from the UEA archive (Appendix A.2.1).

**Implementation details**: In our model, all parameter settings are categorized into three parts: contrastive learning framework parameters, CTFE encoder parameters, and basic training parameters. We set the momentum coefficient $m = 0.92$, the prototype update rate $\gamma = 0.9$, the pseudo target update rate $\phi = 0.9$, the $d_{ci} = 1024$, $d_{feature} = d_{prototype} = 256$, $Batchsize = 512$, $lr = 0.00001$ as the standard hyperparameter setting.

**Experimental design**: We design three main experiments to evaluate the effectiveness of our method:

1. **MTSC with PLL**: In this experiment, we simulate the PLL setting by randomly generating candidate label sets using threshold selection on standard MTSC datasets. For a label set $\gamma = \{y_1, y_2, ...y_N\}, y_i \in \mathcal{Y}, \mathcal{Y} = \{1, 2, 3, ...C\}$, we take the following steps to generate a simulated candidate label set:

   (a) Generate a random vector $r_i \in [0, 1]^C$ from the uniform distribution:
   $$r_i = [r_i^{(1)}, r_i^{(2)}, \ldots, r_i^{(C)}], \quad r_i^{(c)} \sim \mathcal{U}(0, 1)$$

   (b) Define a threshold $\tau$(e.g., based on partial rate), and construct an initial candidate vector $v_i \in \{0, 1\}^C$:
   $$v_i^{(c)} = \begin{cases} 1, & \text{if } r_i^{(c)} < \tau \\ 0, & \text{otherwise} \end{cases}$$

   (c) Ensure the ground-truth label $y_i$ is included in the candidate set by forcing $v_i^{(y_i)} = 1$.

   (d) The candidate label set $\mathcal{S}_i$ is then defined as:
   $$\mathcal{S}_i = \left\{ c \in \mathcal{Y} \,\middle|\, v_i^{(c)} = 1 \right\}$$

   We evaluate model performance under varying partial label rates and compare it with other methods. The analysis includes basic classification accuracy, degradation trends under different partial label rates.

2. **Standard MTSC Benchmark**: This experiment evaluates our model on standard multivariate time series classification (MTSC) tasks without partial labels. We compare the classification accuracy of our method against state-of-the-art baselines to verify its effectiveness in fully supervised scenarios.

3. **Ablation Study**: We conduct ablation experiments by removing key modules from our framework to analyze their contributions. This helps validate the necessity and rationality of the overall design.

**Baselines**: We select some classic baselines and recent advanced MTSC-dedicated models in the MTSC field. Baselines are as follows: **MLSTM-FCN**, **XEM**, **TapNet**, **DA-Net** Chen et al. (2022), **MiniROCKET**, **Conv-GRU** Liu et al. (2023a), **TS2Vec** Yue et al. (2022), **TodyNet** citeliu2024todynet, **MICOS** Hao et al. (2023), **TimesURL** Liu & Chen (2024). **VQShape** Wen et al. (2024).

Table 1: Performance comparison with the recent advanced MTSC models on 20 UEA datasets. In the table, 'N/A' indicates that the results for the corresponding method could not be obtained.

| Dataset IDX | MLSTM-FCN | XEM | Tapnet | DA-net | Mini ROCKET | Conv -GRU | TS2Vec | TodyNet | VQshape | Ours |
|---|---|---|---|---|---|---|---|---|---|---|
| AWR | 0.973 | **0.993** | 0.987 | 0.980 | 0.992 | 0.973 | 0.980 | 0.987 | 0.987 | **0.993** |
| EW | 0.504 | 0.527 | 0.489 | 0.489 | **0.954** | 0.811 | 0.840 | 0.840 | 0.603 | 0.895 |
| EP | 0.761 | 0.986 | 0.971 | 0.883 | **1.000** | 0.978 | 0.942 | 0.971 | 0.893 | 0.986 |
| EC | 0.373 | 0.372 | 0.323 | 0.338 | 0.380 | 0.332 | 0.285 | 0.350 | 0.325 | **0.430** |
| FD | 0.545 | 0.614 | 0.556 | 0.648 | 0.631 | 0.640 | 0.517 | 0.627 | 0.653 | **0.685** |
| FM | 0.580 | 0.590 | 0.530 | 0.510 | 0.450 | 0.580 | 0.540 | 0.570 | **0.642** | 0.600 |
| HW | 0.286 | 0.287 | 0.357 | 0.159 | 0.511 | 0.451 | **0.579** | 0.436 | 0.270 | 0.375 |
| HB | 0.663 | 0.761 | 0.751 | 0.624 | **0.771** | 0.746 | 0.737 | 0.756 | 0.663 | 0.766 |
| IW | NA | 0.228 | 0.208 | 0.567 | 0.595 | 0.208 | 0.179 | NA | NA | **0.700** |
| LIB | 0.856 | 0.772 | 0.850 | 0.800 | 0.878 | 0.889 | 0.867 | 0.850 | 0.814 | **0.900** |
| LSST | 0.373 | 0.652 | 0.568 | 0.560 | 0.643 | 0.548 | 0.545 | 0.615 | 0.511 | **0.676** |
| MI | 0.510 | 0.600 | 0.590 | 0.500 | 0.550 | 0.512 | 0.460 | 0.640 | **0.680** | 0.640 |
| NATOPS | 0.916 | 0.939 | 0.323 | 0.878 | 0.928 | 0.916 | 0.922 | **0.972** | 0.810 | 0.850 |
| PD | 0.978 | 0.977 | 0.980 | 0.980 | 0.965 | 0.939 | 0.981 | **0.987** | 0.973 | **0.987** |
| PM | 0.110 | 0.288 | 0.175 | 0.093 | 0.292 | 0.215 | 0.231 | **0.309** | 0.087 | 0.261 |
| RS | 0.803 | **0.941** | 0.868 | 0.803 | 0.868 | 0.888 | 0.901 | 0.803 | 0.851 | 0.855 |
| SCP1 | 0.874 | 0.839 | 0.652 | **0.924** | 0.874 | 0.843 | 0.741 | 0.898 | 0.904 | **0.924** |
| SCP2 | 0.472 | 0.550 | 0.550 | 0.561 | 0.522 | 0.556 | 0.556 | 0.550 | 0.596 | **0.600** |
| SAD | **0.990** | 0.973 | 0.983 | 0.980 | 0.100 | 0.863 | 0.978 | NA | 0.976 | **0.990** |
| UW | 0.891 | 0.897 | 0.894 | 0.833 | 0.916 | 0.919 | 0.913 | 0.850 | 0.888 | **0.934** |
| Total best accuracy | 1 | 2 | 0 | 1 | 3 | 0 | 1 | 3 | 2 | **11** |
| Total second accuracy | 0 | 3 | 1 | 0 | **5** | 2 | 2 | 1 | 3 | 5 |
| Ours 1-to-1-Wins | 18 | 15 | 19 | 19 | 13 | 17 | 16 | 15 | 18 | - |
| Ours 1-to-1-Draws | 1 | 2 | 0 | 0 | 0 | 0 | 1 | 2 | 0 | - |
| Ours 1-to-1-Losses | 1 | 3 | 1 | 1 | 7 | 3 | 3 | 3 | 2 | - |
| Avg.ACC all($\uparrow$) | 0.623 | 0.689 | 0.630 | 0.656 | 0.691 | 0.690 | 0.685 | 0.651 | 0.656 | **0.752** |
| Avg.ACC w/o IW,SAD($\uparrow$) | 0.637 | 0.699 | 0.634 | 0.642 | 0.729 | 0.708 | 0.697 | 0.723 | 0.675 | **0.742** |

Table 2: Ablation study on the modules: DMDA,CI-Encoder,MiniROCKET-Encoder,FusionGate.

| Dataset | Setting | | | | Accuracy | | |
|---|---|---|---|---|---|---|---|
| | DMDA | CI Encoder | MiniROCKET Encoder | Fusion Gate | partial 0 | partial 0.25 | partial 0.5 |
| AWR | | | | | **0.983** | **0.967** | **0.527** |
| | ✗ | | | | 0.981 | 0.807 | 0.187 |
| | | ✗ | | | 0.967 | 0.920 | 0.290 |
| | | | ✗ | | 0.977 | 0.912 | 0.350 |
| | | | | ✗ | 0.970 | 0.933 | 0.263 |
| IW | | | | | **0.671** | **0.652** | **0.601** |
| | ✗ | | | | 0.669 | 0.647 | 0.584 |
| | | ✗ | | | 0.658 | 0.643 | 0.594 |
| | | | ✗ | | 0.577 | 0.424 | 0.230 |
| | | | | ✗ | 0.641 | 0.629 | 0.572 |

## 4.2 MTSC-PLL RESULTS

We set Partial-rate=$\{0, 0.125, 0.25, 0.375, 0.5, 0.625\}$to compare the performance of our model and other models under the PLL condition. We use standard hyperparameter settings for the models used for comparison (including CO-COME). Figure 3 illustrates the results of 6 datasets (see the Appendix2.3 for results on other datasets): cross all evaluated datasets (AWR, IW, LIB, NAT, SAD, UW), CO-COME consistently achieves the highest or near-highest accuracy when the partial rate increases. In low label ambiguity scenarios (partial-rate=0), our performance is comparable to other strong baselines (e.g., MiniROCKET, MICOS). When the partial-rate grows, competing methods experience steep performance degradation. By contrast, CO-COME shows markedly slower accuracy decay and remains stable even at partial-rate=0.5. For example, on SAD and UW, CO-COME retains above 0.95 and 0.68 at partial-rate=0.625, clearly outperforming others. These results demonstrate that CO-COME provides robust classification under severe label ambiguity, validating the effectiveness of its design for partial-label time series classification.

## 4.3 MTSC-BASIC RESULTS

Our method consistently demonstrates superior performance across the 20 UEA standard datasets compared to all the baseline in Table 2. Our method achieves the highest average accuracy of 0.752 across all datasets, outperforming all compared baselines. Even when excluding datasets where some baselines lack reported results (IW and SAD), our approach maintains leading performance with an average accuracy of 0.742. In terms of per-dataset comparison, our method achieves the highest number of first-place scores 11, significantly ahead of other models. In head-to-head comparisons, our method records 15+ wins against most baselines, with minimal losses. Particularly, these results indicate the robustness and generalization ability of our approach across diverse time series classification tasks.

## 4.4 ABLATION STUDIES

To evaluate the effectiveness of different components, we conduct ablation experiments on two datasets with 4 variants under three different partial label conditions. Table 2 illustrates the results: compared to the base line, removing any component leads to a decrease in accuracy, especially when the partial rate increases. Ablation results clearly verify that each component contributes to the overall robustness of our framework. The performance gaps become larger as the partial label ratio grows, indicating that DMDA, CI-Encoder, MiniROCKET-Encoder, FusionGate, and CTFE are all necessary for maintaining stable performance.

## 5 CONCLUSION

Experimental results demonstrate that our proposed CO-COME model achieves state-of-the-art performance on multivariate time series classification (MTSC) under both standard and partial-label learning (PLL) conditions. This work effectively fills the gap in MTSC under the PLL condition and offers a practical solution for real-world scenarios.

Beyond benchmarks, our framework can be used in industrial pipelines to mitigate label noise by reframing it as a partial-label problem and training with CO-COME. Noisy or conflicting annotations are converted into candidate label sets via confidence thresholds, annotator-agreement filters, taxonomy coarsening, or top-k model scores. CO-COME then disambiguates labels through learning robust representations, which reduce relabeling cost and improve reliability on real-world multivariate time series.

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

## A APPENDIX

### A.1 DMDA

Excessive or insufficient data augmentation can impair the effectiveness of contrastive learning by reducing its representation quality. Moreover, aggressive augmentation may shift the data distribution between the training and test sets, causing a significant discrepancy in the fixed features extracted by the MiniROCKET module across these domains.

The DMDA module analyzes inter-variable correlations across the entire MTS dataset and computes key statistical indicators, including the average, maximum, and minimum correlation coefficients. These metrics are used to guide the selection and strength of augmentation strategies.

Based on the correlation analysis, we apply a set of tailored data augmentation strategies. The selected augmentation techniques include:

- **Time Mask**: This method randomly masks a small portion of the time steps within each channel. Given a multivariate time series $X \in \mathbb{R}^{T \times M}$, we define the masked index set as:

$$\mathcal{S} = \{(t_i, m_i) \mid t_i \in \{1, \ldots, T\}, \ m_i \in \{1, \ldots, M\}\}$$
$$|\mathcal{S}| = \lfloor \sigma \cdot T \cdot M \rfloor$$

  Then, we apply the mask:

$$X_{t,m} = \begin{cases} \varepsilon, & \text{if } (t,m) \in \mathcal{S} \\ X_{t,m}, & \text{otherwise} \end{cases} \quad \text{with } \varepsilon \approx 10^{-8}, \ \varepsilon > 0$$

- **Time Warp**: This method applies a deterministic sinusoidal perturbation to each channel, introducing smooth, non-linear temporal distortion. For each channel $m$, we generate a sinusoidal vector of length $T$ and apply it with scaling coefficient $\alpha$:

$$\delta(t) = \sin\left(\frac{t}{T} \cdot \pi\right), \quad t \in \{1, \ldots, T\}$$
$$X_{t,m}^{\text{warp}} = X_{t,m} + \alpha \cdot \delta(t)$$

  where $\alpha$ controls the warping strength. This operation preserves the signal structure while applying smooth temporal modulation.

- **Scaling**: This method perturbs the amplitude of each channel by scaling factor $\beta$. For each channel $m$, we sample a scaling coefficient:

$$s_m \sim \mathcal{U}(1 - \beta, \ 1 + \beta)$$

  and apply it to all time steps in the channel:

$$X_{t,m}^{\text{scale}} = s_m \cdot X_{t,m}, \quad \forall t \in \{1, \ldots, T\}$$

- **Frequency Perturbation**: This method perturbs the frequency components of the time series via the Fast Fourier Transform (FFT). We transform each channel $m$ into the frequency domain:

$$F_m = \text{FFT}(X_{:,m})$$

  Then add random Gaussian noise in the frequency domain:

$$\tilde{F}_m = F_m + \eta_m, \quad \eta_m \sim \mathcal{N}(0, \lambda^2)$$

  Finally, we inverse transform to obtain the perturbed signal:

$$X_{:,m}^{\text{freq}} = \text{IFFT}(\tilde{F}_m)$$

### A.2 EXPERIMENTAL SETTINGS

#### A.2.1 DATASETS

The public UEA benchmark datasets, collected from various real-world applications, constitute a comprehensive archive for MTSC across several domains, including EEG, insects, speech, human activity, and audio data, etc. We remove the dataset with excessively high dimensionality or insufficient training data. Table 3 summarizes the selected datasets, including index(IDX), dataset name, number of classes, training and test set sizes, series length, and number of dimensions.

Table 3: Detail information of the selected 20 UEA datasets

| Dataset Index | Dataset Name | Num Classes | Train Size | Series Length | Test Size | Num Dimensions | Type |
|---|---|---|---|---|---|---|---|
| AWR | ArticularyWordRecognition | 25 | 275 | 144 | 300 | 9 | Motion |
| EW | EigenWorms | 5 | 128 | 17984 | 131 | 6 | Motion |
| EP | Epilepsy | 4 | 137 | 206 | 138 | 3 | HAR |
| EC | EthanolConcentration | 4 | 261 | 1751 | 263 | 3 | HAR |
| FD | FaceDetection | 2 | 5890 | 62 | 3524 | 144 | EEG/MEG |
| FM | FingerMovements | 2 | 316 | 50 | 100 | 28 | EEG/MEG |
| HW | Handwriting | 26 | 150 | 152 | 850 | 3 | HAR |
| HB | Heartbeat | 2 | 204 | 405 | 205 | 61 | AS |
| IW | InsectWingbeat | 10 | 30000 | 30 | 20000 | 200 | AS |
| LIB | Libras | 15 | 180 | 45 | 180 | 2 | HAR |
| LSST | LSST | 14 | 2459 | 36 | 2466 | 6 | Others |
| MI | MotorImagery | 2 | 278 | 3000 | 100 | 64 | EEG/MEG |
| NAT | NATOPS | 6 | 180 | 51 | 180 | 24 | HAR |
| PD | PenDigits | 10 | 7494 | 8 | 3498 | 2 | EEG/MEG |
| PS | Phoneme | 39 | 3315 | 217 | 3353 | 11 | AS |
| RS | RacketSports | 4 | 151 | 30 | 152 | 6 | HAR |
| SCP1 | SelfRegulationSCP1 | 2 | 268 | 896 | 293 | 6 | EEG/MEG |
| SCP2 | SelfRegulationSCP2 | 2 | 200 | 1152 | 180 | 7 | EEG/MEG |
| SAD | SpokenArabicDigits | 10 | 6599 | 93 | 2199 | 13 | AS |
| UW | UWaveGestureLibrary | 8 | 120 | 315 | 320 | 3 | HAR |

### A.2.2 IMPLEMENTATION DETAILS

All comparative methods are reproduced using their official code implementations and settings. Furthermore, each experiment is conducted on a workstation with an NVIDIA H20 GPU. We set the momentum coefficient $m = 0.92$, the prototype update rate $\gamma = 0.9$, the pseudo target update rate $\phi = 0.9$, the loss weight, $Batchsize = 512$, $lr = 0.00001$. Keeping the above parameters, we vary the CTFE parameters $d_{ci}$ and $d_{feat}$ across five configurations:

$$A : d_{ci} = 256, d_{feat} = 64$$

$$B : d_{ci} = 512, d_{feat} = 64$$

$$C : d_{ci} = 512, d_{feat} = 128$$

$$D : d_{ci} = 1024, d_{feat} = 128$$

$$E : d_{ci} = 1024, d_{feat} = 256$$

As shown in Table-4, small $d_{ci}$ and $d_{feat}$ yield lower accuracy and weaker robustness as the partial rate increases. With $d_{ci} = 1024$, overall accuracy is almost strong across datasets. Moreover, at $d_{ci} = 1024$, choosing $d_{feat} = 256$ surpasses $d_{feat} = 128$ by delivering higher aggregate accuracy and slower accuracy decay at larger partial rates. Particularly, although the IW dataset achieves its best performance with $d_{ci} = 256$ and $d_{feat} = 64$, we adopt $d_{ci} = 1024$ and $d_{feat} = 256$ for subsequent experiments to optimize overall accuracy and robustness across datasets.

### A.2.3 MTSC-PLL RESULTS

We conduct PLL experiments on 15 datasets, Figure4 illustrates the results: Across 11 datasets, when partial-rate reaches 0.625, CO-COME outperforms all baselines by large margins. On nine datasets(AWR, IW, LIB, NAT, SAD, UW, EW, FD, RS), the gap emerges and persists when partial-rate reaches 0.375. By contrast, on HW and LSST all methods exhibit a similar downward trend with increasing partial-rate, and accuracies drop to low levels. These two datasets contain $> 10$ classes. We attribute this to the inherent difficulty of these tasks. For HB and SCP2, accuracies are largely insensitive to partial-rate. This means that the relationship between labels and data in these two datasets is not that strong, the datasets themselves may have certain defects. Overall, CO-COME maintains strong performance and robustness at high partial-rate, supporting its effectiveness in disambiguating partial labels.

Table 4: Accuracy across partial rates for four datasets under four hyperparameter settings.

| Dataset | partial-rate | $d_{ci}$=256 $d_{feat}$=64 | $d_{ci}$=512 $d_{feat}$=64 | $d_{ci}$=512 $d_{feat}$=128 | $d_{ci}$=1024 $d_{feat}$=128 | $d_{ci}$=1024 $d_{feat}$=256 |
|---|---|---|---|---|---|---|
| | 0.0000 | 0.777 | 0.787 | 0.890 | 0.980 | **0.987** |
| | 0.0625 | 0.803 | 0.820 | 0.917 | 0.983 | **0.990** |
| | 0.1250 | 0.743 | 0.707 | 0.800 | 0.987 | **0.990** |
| AWR | 0.1875 | 0.610 | 0.543 | 0.613 | **0.983** | **0.983** |
| | 0.2500 | 0.580 | 0.467 | 0.570 | **0.987** | 0.967 |
| | 0.3125 | 0.533 | 0.403 | 0.350 | 0.893 | **0.983** |
| | 0.3750 | 0.303 | 0.233 | 0.223 | 0.827 | **0.837** |
| | 0.4375 | 0.213 | 0.173 | 0.203 | **0.707** | 0.683 |
| $\Delta$ (0→0.4375) | | 0.564 | 0.614 | 0.687 | 0.273 | 0.304 |
| | 0.0000 | **0.678** | 0.664 | 0.666 | 0.663 | 0.655 |
| | 0.0625 | **0.676** | 0.656 | 0.664 | 0.653 | 0.652 |
| | 0.1250 | **0.674** | 0.658 | 0.661 | 0.643 | 0.645 |
| IW | 0.1875 | **0.669** | 0.655 | 0.657 | 0.646 | 0.645 |
| | 0.2500 | **0.666** | 0.650 | 0.651 | 0.646 | 0.645 |
| | 0.3125 | **0.666** | 0.646 | 0.640 | 0.640 | 0.632 |
| | 0.3750 | **0.653** | 0.635 | 0.635 | 0.630 | 0.631 |
| | 0.4375 | **0.645** | 0.627 | 0.618 | 0.600 | 0.609 |
| $\Delta$ (0→0.4375) | | 0.033 | 0.037 | 0.048 | 0.063 | 0.046 |
| | 0.0000 | 0.628 | 0.733 | 0.700 | **0.900** | 0.894 |
| | 0.0625 | 0.656 | 0.689 | 0.672 | 0.861 | **0.883** |
| | 0.1250 | 0.667 | 0.639 | 0.650 | **0.833** | 0.817 |
| LIB | 0.1875 | 0.661 | 0.611 | 0.661 | **0.839** | **0.839** |
| | 0.2500 | 0.639 | 0.600 | 0.606 | **0.800** | **0.800** |
| | 0.3125 | 0.622 | 0.517 | 0.422 | **0.783** | 0.778 |
| | 0.3750 | 0.567 | 0.389 | 0.394 | **0.722** | 0.706 |
| | 0.4375 | 0.500 | 0.211 | 0.272 | 0.500 | **0.672** |
| $\Delta$ (0→0.4375) | | 0.128 | 0.522 | 0.428 | 0.400 | 0.222 |
| | 0.0000 | 0.778 | 0.800 | 0.794 | 0.828 | **0.833** |
| | 0.0625 | 0.733 | 0.772 | 0.811 | **0.839** | 0.822 |
| | 0.1250 | 0.772 | 0.739 | 0.772 | **0.806** | 0.794 |
| NAT | 0.1875 | 0.772 | 0.750 | 0.778 | **0.800** | **0.800** |
| | 0.2500 | 0.722 | 0.722 | 0.772 | **0.794** | 0.789 |
| | 0.3125 | 0.717 | 0.744 | 0.773 | 0.806 | **0.833** |
| | 0.3750 | 0.717 | 0.739 | 0.750 | 0.800 | **0.828** |
| | 0.4375 | 0.711 | 0.711 | 0.778 | 0.789 | **0.794** |
| $\Delta$ (0→0.4375) | | 0.067 | 0.089 | 0.016 | 0.039 | 0.039 |
| | 0.0000 | 0.975 | 0.973 | 0.972 | 0.982 | **0.985** |
| | 0.0625 | 0.977 | 0.975 | 0.974 | 0.982 | **0.986** |
| | 0.1250 | 0.974 | 0.971 | 0.972 | **0.983** | **0.983** |
| SAD | 0.1875 | 0.972 | 0.972 | 0.970 | 0.976 | **0.978** |
| | 0.2500 | 0.974 | 0.970 | 0.967 | **0.977** | **0.977** |
| | 0.3125 | 0.969 | 0.964 | 0.968 | **0.977** | 0.975 |
| | 0.3750 | 0.970 | 0.964 | 0.962 | **0.972** | 0.970 |
| | 0.4375 | 0.965 | 0.963 | 0.960 | **0.971** | **0.971** |
| $\Delta$ (0→0.4375) | | 0.010 | 0.010 | 0.012 | 0.011 | 0.014 |
| | 0.0000 | 0.800 | 0.853 | 0.825 | **0.931** | 0.916 |
| | 0.0625 | 0.794 | 0.863 | 0.831 | **0.925** | 0.906 |
| | 0.1250 | 0.778 | 0.872 | 0.825 | **0.947** | 0.910 |
| UW | 0.1875 | 0.769 | 0.834 | 0.841 | **0.925** | 0.910 |
| | 0.2500 | 0.763 | 0.828 | 0.844 | 0.900 | **0.906** |
| | 0.3125 | 0.734 | 0.784 | 0.806 | 0.888 | **0.916** |
| | 0.3750 | 0.559 | 0.816 | 0.813 | **0.919** | 0.891 |
| | 0.4375 | 0.550 | 0.797 | 0.750 | 0.875 | **0.890** |
| $\Delta$ (0→0.4375) | | 0.250 | 0.056 | 0.075 | 0.056 | 0.026 |
| **Best count** | | 8 | 0 | 0 | 23 | 24 |

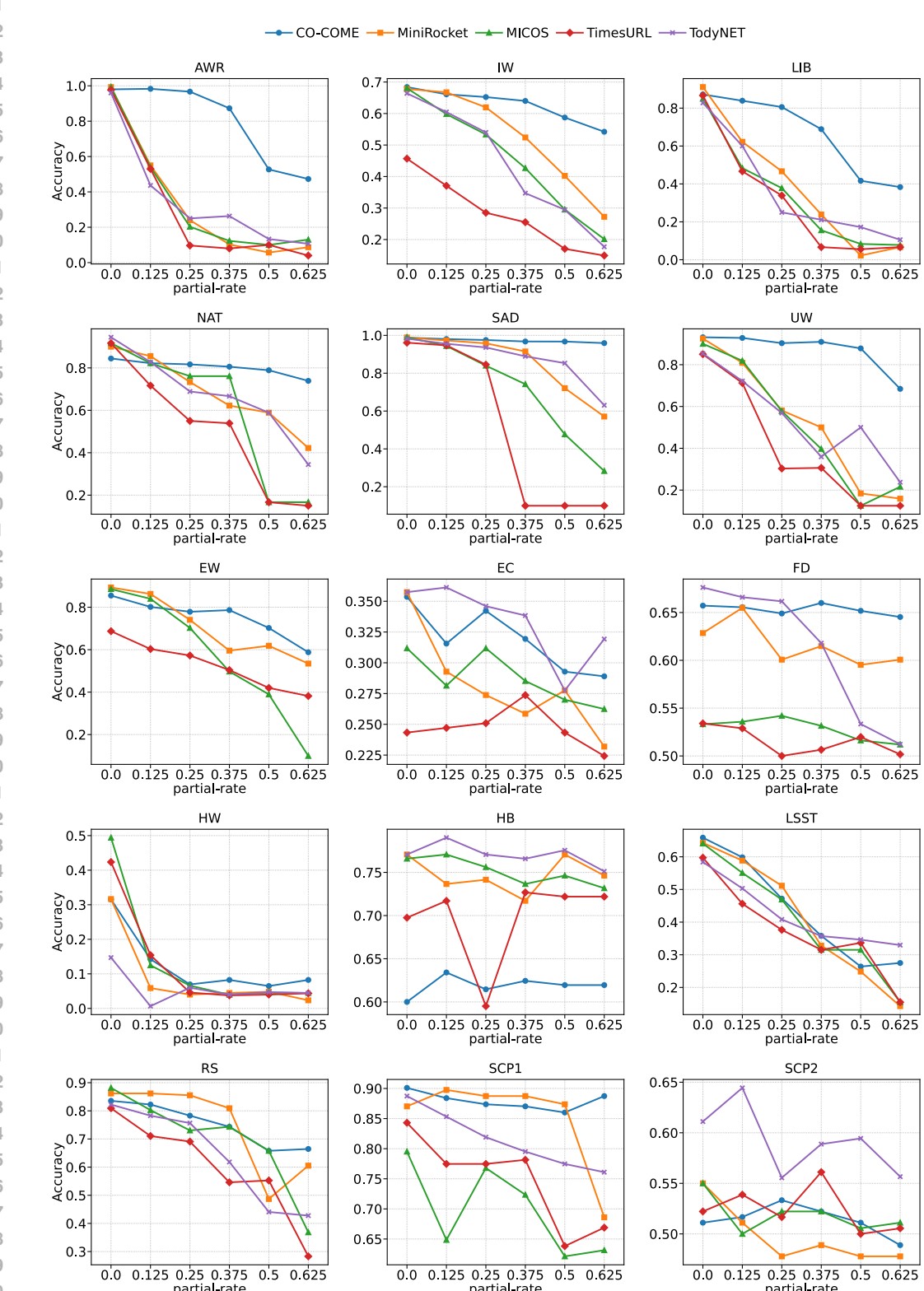

Figure 4: Accuracy on 15 datasets under PLL condition.

### A.2.4 T-SNE VISUALIZATION OF WEAK/STRONG AUGMENTED FEATURES

We further visualize the weak and strong augmented features $f_W$ and $f_S$ using t-SNE, as shown in Fig.5. Across training, both views gradually form clearer class-related structure, indicating that the proposed framework learns meaningful and discriminative representations with stable convergence. The weak and strong augmentations yield effective embeddings

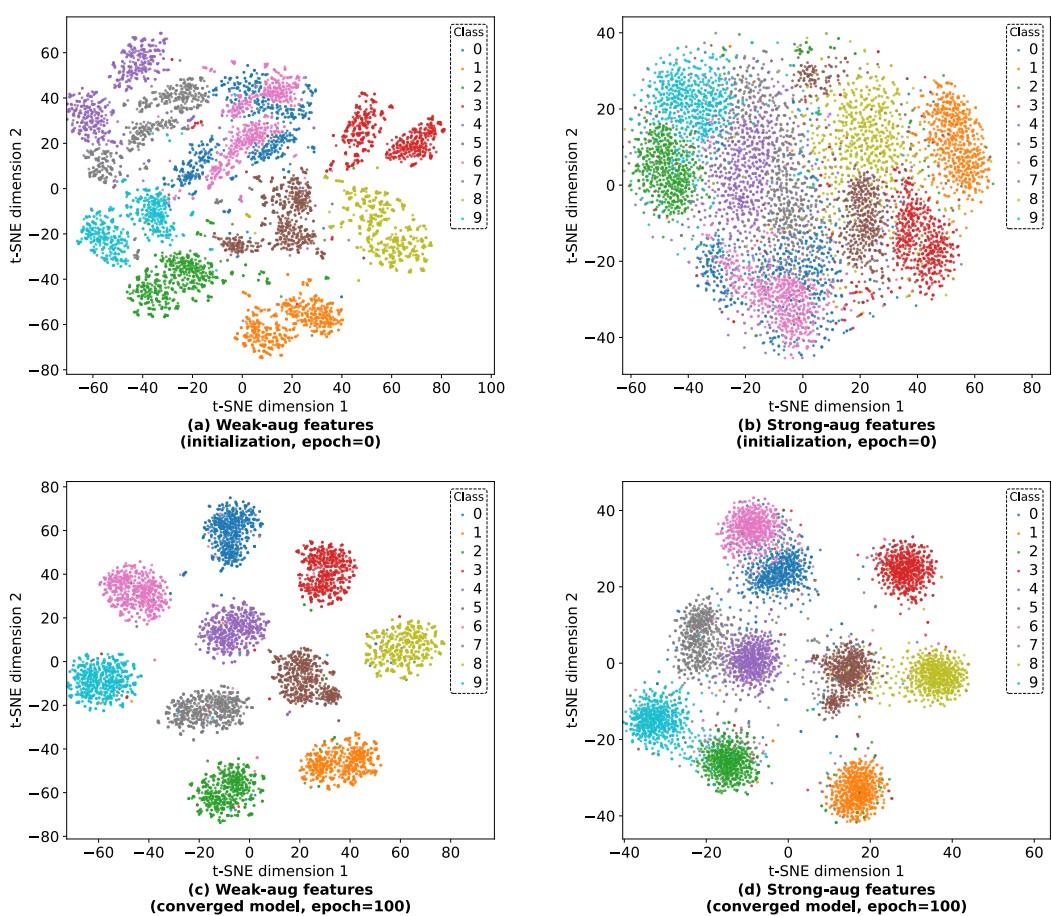

Figure 5: T-SNE Visualization of the CTFE representation on SAD with partail-rate=0.5

