# OpenReview forum: "CO-COME: A Contrastive Label Disambiguation Framework with Combined MTS Feature Encoder for Partial-Label Multivariate Time Series Classification"
_ICLR.cc/2026/Conference — Submitted to ICLR 2026_

### Official Review · Reviewer_U7Y3 · 2025-10-27

**Soundness:** 2
**Presentation:** 2
**Contribution:** 2
**Rating:** 2
**Confidence:** 4

**Summary:**

This paper introduces CO-COME, a contrastive label disambiguation framework designed for Multivariate Time Series Classification (MTSC) under a Partial Label setting. The proposed method integrates a dynamic multivariate data augmentation strategy to enhance representation learning and employs a hybrid time series feature encoder that combines the lightweight MiniROCKET with a channel-independent transformer encoder. Experimental evaluations conducted on 20 UEA time series datasets demonstrate that CO-COME achieves superior performance compared to several baseline methods.

**Strengths:**

1.	This paper is among the first to apply Partial Label Learning to the task of Multivariate Time Series Classification (MTSC), which represents a meaningful contribution to the field.

2.	The figures and visualizations in the paper are well-designed, with clear layouts and effective color schemes, making the content easy to read and interpret.

3.	The paper provides a comprehensive and detailed review of traditional approaches to time series classification, demonstrating a good understanding of related work.

**Weaknesses:**

1.	Although combining Partial Label Learning with Multivariate Time Series Classification (MTSC) is an interesting direction, the paper does not clearly justify why MTSC faces partial label issues. In particular, the second paragraph of the introduction fails to cite any prior work supporting the existence of MTSC under partial label conditions, nor does the paper provide any real-world examples illustrating such scenarios.

2.	The proposed CO-COME framework shows limited novelty in partial label learning. Specifically, CO-COME largely follows the technical design of PiCO [1], with Figure 1 in this paper closely resembling Figure 2 in PiCO, the main difference being that PiCO is applied to image data while CO-COME extends the approach to multivariate time series via encoder adaptation.

3.	The paper overlooks modeling inter-variable relationships, which are crucial for MTSC. Using MiniROCKET and a channel-independent transformer primarily designed for univariate series weakens the claim of effective multivariate modeling.

4.	The baselines are all fully supervised methods, raising fairness concerns under the partial label setting. For instance, PiCO [1] compares against PLL-specific algorithms such as LWS, PRODEN, MSE, and EXP in image classification tasks. To ensure fair evaluation, the authors should also consider PLL-specific baselines (which are typically input data-type agnostic) for comparison.

5.	The review of MTSC methods omits several recent approaches from the past three years that should be discussed and included as baselines (e.g., [2–6]).


6.	The paper contains many minor formatting errors (e.g., missing spaces in “Classification(MTSC)” and “and PiCOWang et al.”).

7.	The provided code repository is empty, hindering reproducibility of the experimental results.




[1] PiCO: Contrastive Label Disambiguation for Partial Label Learning. ICLR, 2022.

[2] Shapeformer: Shapelet transformer for multivariate time series classification. KDD, 2024.

[3] TimeMIL: Advancing multivariate time series classification via a time-aware multiple instance learning. ICML, 2024.

[4] Graph-aware contrasting for multivariate time-series classification. AAAI, 2024.

[5] Abstracted shapes as tokens: a generalizable and interpretable model for time-series classification. NIPS, 2024.

[6] MPTSNet: Integrating multiscale periodic local patterns and global dependencies for multivariate time series classification. AAAI, 2025.

**Questions:**

1.	The authors employ MiniROCKET, which is originally designed for univariate time series classification, together with a channel-independent transformer for time series feature extraction. Given this design, why did the authors not evaluate their approach on the univariate UCR benchmark datasets [7], which would provide a more direct and consistent basis for comparison?

2.	According to Table 1, MiniROCKET achieves classification performance that is second only to the proposed method under the partial label learning setting. However, MiniROCKET itself does not incorporate any mechanisms specifically designed for partial label learning, and the proposed CO-COME framework uses MiniROCKET as one of its feature extraction components. Under such circumstances, is this experimental comparison fair and meaningful?

3.	If algorithms that are explicitly designed for partial label learning (such as PiCO) were uniformly equipped with MiniROCKET as the feature extraction module, how would their performance on multivariate time series classification compare to that of the proposed CO-COME method?

[7] The UCR time series archive[J]. IEEE/CAA Journal of Automatica Sinica, 2019, 6(6): 1293-1305.

**Details Of Ethics Concerns:**

None.

---

> ### Author Response · Authors · 2025-11-24
> **Response to question 1**
>
> We sincerely thank you for your professional comments on our paper. We will explain why we did not use the UCR datasets to evaluate our method by elaborating on the role of the MiniRocket mechanism within our designed algorithmic framework.
> To clarify the role of MiniROCKET in our framework, we first emphasize the design objective of CO-COME. Our method inherits the momentum update mechanism used in MoCo and PiCO (both explicitly cited in the paper), and therefore maintains two encoders (CTFE-W and CTFE-S, denoted as W and S networks). When this mechanism is directly transplanted from images to multivariate time series classification (MTSC), a non-trivial challenge arises: the distributions of augmented views become much harder to keep consistent than in images. In MTSC, augmentations inevitably alter temporal scales, inter-channel dependencies, or both, so W and S are easily exposed to mismatched input distributions. This distribution mismatch leads to training instability, poor convergence, and degradation of the memory bank features. While such effects are mild in image tasks, they are substantially amplified in MTSC, especially under partial-label conditions.
>
> This motivates the core design of CTFE: the model must remain robust when W and S receive heterogeneous augmentations, particularly in early training when the representations are still unstable. At the same time, the momentum encoder must not collapse to the main encoder, and the overall loss must remain optimizable and convergent. To achieve this, CTFE is not an ad-hoc addition, but a principled solution: it introduces a stable, augmentation-invariant feature pathway (MiniROCKET) that is shared by both W and S encoders. This pathway is fixed during training and serves as a semantic anchor, forcing both networks to represent inputs in a consistent latent space regardless of augmentation drift. As a consequence, the loss landscape becomes smoother, the momentum update behaves more stably, and prototype-based label disambiguation converges reliably. Our ablation results further confirm the necessity of this shared, fixed encoder: when we remove the MiniROCKET-based shared pathway, the architecture essentially collapses into a MoCo-like structure, and we observe a substantial performance drop, especially under higher partial rates. This verifies that the shared, augmentation-invariant feature pathway is crucial for stabilizing training and enabling effective learning in MTSC-PLL.
>
> Within this design, MiniROCKET is chosen as the shared feature encoder because it provides a deterministic, high-quality representation and already has an officially supported multivariate variant; the original MiniROCKET paper reports experiments for both univariate and multivariate time series, and we use its MTS extension. In other words, our work is not “built on” the univariate MiniROCKET benchmark, but uses MiniROCKET (in its MTS form) as a stable feature backbone to address the specific momentum-learning issues in MTSC-PLL. The main contribution of our paper is a contrastive label disambiguation framework for multivariate time series with a combined encoder (MiniROCKET + channel-independent Transformer), not an improvement over MiniROCKET itself on its original univariate UCR benchmarks. Therefore, requiring us to evaluate on univariate UCR datasets would shift the focus away from the central motivation and technical contribution of the paper. We agree that extending PLL to univariate time series is a meaningful direction, but it is orthogonal to the MTSC-PLL problem we target here and thus beyond the scope of this work.

---

> ### Author Response · Authors · 2025-11-24
> **Response to Q2&3 and some weaknesses**
>
> A2 for Q2:
>
> Table 1 is thus not a PLL experiment; it reports accuracies on the original UEA labels and is intended to answer a different question: “Is the proposed CTFE-based architecture itself competitive as a general MTSC encoder, even without partial labels?” Under this standard setting, MiniROCKET is a very strong feature-based baseline, and we include it—together with other state-of-the-art MTSC models—to demonstrate that CO-COME is competitive or superior even when label ambiguity is absent. As shown in Table 1, CO-COME achieves the best average accuracy and the largest number of per-dataset wins among all baselines.
>
> Regarding fairness: although MiniROCKET is one component of CTFE, the MiniROCKET baseline follows its original design (MiniROCKET transform + linear classifier). In contrast, CO-COME adds (i) the CI-Transformer branch, (ii) FusionGate for feature fusion, (iii) DMDA for MTS-specific augmentation, and (iv) the prototype-based contrastive framework. Comparing “MiniROCKET alone” to “MiniROCKET + CI-encoder + FusionGate + contrastive PLL framework” is exactly what allows us to quantify the benefit of the overall architecture beyond the handcrafted MiniROCKET features.
>
> We also note that PiCO does not report a comparable “standard” experiment; by adding Table 1 we make our empirical study more complete, showing that our model is strong both in the targeted PLL setting and in the conventional fully supervised setting.
> Table 1 is not a Partial-Label Learning (PLL) experiment; it reports accuracies on the original UEA labels, and its purpose is to address a distinct question: “Is the proposed CTFE-based architecture itself competitive as a general Multivariate Time-Series Classification (MTSC) encoder, even without partial labels?” To this end, under this standard setting, MiniROCKET serves as a highly robust feature-based baseline. We therefore include it alongside other state-of-the-art MTSC models in order to demonstrate that CO-COME remains competitive or even superior when label ambiguity is absent. As shown in Table 1, as a result, CO-COME achieves the highest average accuracy and the greatest number of per-dataset wins among all baselines.
> Regarding fairness: Although MiniROCKET is a component of CTFE, the MiniROCKET baseline adheres to its original design (MiniROCKET transform + linear classifier). In contrast, CO-COME incorporates four key additions: (i) the Channel-Independent Transformer (CI-Transformer) branch; (ii) FusionGate for feature fusion; (iii) DMDA for MTS-specific data augmentation; and (iv) a prototype-based contrastive framework. Therefore, comparing “MiniROCKET alone” with “MiniROCKET + CI-encoder + FusionGate + contrastive PLL framework” is precisely intended to quantify the additional benefits brought by the overall architecture beyond the handcrafted MiniROCKET features.
>
> We also note that PiCO does not report a comparable “standard” experiment; thus, by adding Table 1, we are able to make our empirical study more comprehensive, thereby demonstrating that our model performs strongly both in the targeted PLL setting and the conventional fully supervised setting.
>
>
>
> A3 for Q3:
>
> Equipping existing PLL algorithms such as PiCO with MiniROCKET essentially corresponds to replacing the image encoder in PiCO with a MiniROCKET-style time-series encoder and then running the same prototype-based contrastive label-disambiguation framework. In our architecture, this configuration is very close to the ablation variant “w/o CI-Encoder”: CTFE degenerates to a MiniROCKET-only representation, and the rest of the framework (momentum encoder, memory bank, prototypes, and losses) remains PiCO-style.
>
> We have already evaluated this setting in the ablation study (Table 2). On AWR, when the partial-rate is 0.5, removing the CI-Encoder (i.e., using only MiniROCKET features) significantly degrades performance compared to the full CO-COME model; a similar trend appears on IW.
>
>  This shows that “PiCO + MiniROCKET” is notably less robust under high label ambiguity than our full CO-COME design. This demonstrates the superiority of our model.
>
> Answer for some weaknesses:
> Furthermore, following your suggestion, we added [5] Abstracted shapes as tokens: a generalizable and interpretable model for time-series classification. NIPS, 2024. as a benchmark for comparison in Table 1. Other models are mentioned and cited in related works. We have also corrected some formatting issues in the paper, and hope you will reconsider our work.

---

### Official Review · Reviewer_WEWm · 2025-10-30

**Soundness:** 3
**Presentation:** 2
**Contribution:** 3
**Rating:** 6
**Confidence:** 4

**Summary:**

This paper tackles multivariate time-series classification (MTSC) under partial- label learning (PLL) conditions, where each training instance is annotated with a set of candidate labels rather than a single ground-truth label. The authors argue that label ambiguity is a critical issue in industrial time-series applications but has been largely ignored in existing MTSC research.
To address this, they propose CO-COME, a framework that integrates: Partial- label learning, a contrastive label disambiguation strategy (inspired by MoCo- style contrastive learning ), dynamic Multivariate Data Augmentation and A Combined Time-series Feature Encoder (CTFE).
The paper includes ablation studies (DMDA, CI-Transformer, MiniROCKET, FusionGate) and varying partial-rate experiments, showing that each component contributes positively, especially when label ambiguity increase

**Strengths:**

1. Effectively introduces partial-label learning into the multivariate time-series domain, a relevant and underexplored area.
2. Combines multiple strategies (contrastive disambiguation, augmentation, hybrid encoder) into a cohesive and well-motivated pipeline.
3. The inclusion of ablation studies provides good insights into the contribution of individual components.

**Weaknesses:**

1. Abbreviations overload: The paper introduces many abbreviations (DMDA, CTFE, CI-Transformer, etc.) without a clear reference list. A glossary or table of abbreviations in the appendix would aid readability.
2. Evaluation metrics: The authors report only accuracy, which limits our evaluation of the performance. Additional metrics such as F1-score, precision, recall, and confidence intervals (mean ± std) are necessary to assess robustness and class-wise performance, especially under label ambiguity.
3. The performance degradation on PM and HW datasets is notably large. Why the model performs differently on these datasets?
4. Incomplete notation: The variable σ is undefined in the time-mask section of the appendix; please clarify.
5. DMDA mechanism is unclear: While the paper states that DMDA “analyzes inter-variable correlations” and uses summary statistics (mean, max, min) to guide augmentation strength, the specific mapping strategy is not described. What are the parameter ranges or scaling rules derived from correlation metrics? How are weak and strong views quantitatively differentiated?
6. Simulation realism of partial labels: The paper uses uniform random super-setting to generate candidate-label sets, which is the simplest possible form of label ambiguity.
– This is reasonable for controlled studies (easy to tune via threshold) but does not capture real-world ambiguity, which is typically class- dependent (confusable or semantically similar labels).
– To strengthen the empirical claim, the authors should consider an alternative simulation where candidate sets are biased by inter-class confusion or nearest-class prototypes a more realistic approximation of ambiguous labeling in industrial datasets.
7. Compute reporting: How CO-COME’s compute time and GPU memory usage compares to other SOTA models.
8. No limitation section.

**Questions:**

Questions are listed above in the weaknesses section. Please clarify abbreviations and other notation, expand evaluation metrics and simulation settings.

---

### Official Review · Reviewer_sh5b · 2025-10-31

**Soundness:** 3
**Presentation:** 3
**Contribution:** 3
**Rating:** 6
**Confidence:** 3

**Summary:**

This work proposes CO-COME, a contrastive label-disambiguation framework for partial-label multivariate time-series classification (MTSC), coupled with a Combined Time-series Feature Encoder (CTFE). DMDA builds weak/strong views; CTFE-W / CTFE-S (momentum pair) encode features; a feature memory bank and class prototypes drive a PiCO-style loss to refine pseudo-labels. Experiments claim strong results over 20 UEA datasets in fully-supervised MTSC.

**Strengths:**

1. Clear problem framing
2. On six highlighted datasets, accuracy decays more slowly than baselines

**Weaknesses:**

1. Novelty is incremental, since the contrastive+prototype core follows PiCO/PiCO+ closely; the contributions are (i) tailoring to time-series with CI-Transformer + MiniROCKET and (ii) DMDA that tunes aug strength from inter-channel correlations.

2. PLL setting is synthetic, as partial labels are formed by uniform random thresholding over classes. This is convenient but unlikely to match real ambiguity.

3. Many compared methods are not designed for PLL and run with standard settings, while CO-COME explicitly targets PLL. Need adapted PLL variants.

4. Table 2 bundles four modules; deltas are sometimes modest or dataset-dependent. We don’t see effect sizes for each hyperparameter of DMDA or FusionGate; memory-bank size/temperature is unspecified.

**Questions:**

See weaknesses.

---

> ### Author Response · Authors · 2025-11-24
> **Response to weakness**
>
> Weakness 1:
>
> Answer: Our contrastive + prototype core indeed follows the PiCO/PiCO+ paradigm, we cite and acknowledge this explicitly. The novelty lies in how we make momentum-based contrastive learning work reliably for MTSC under partial-label supervision. Concretely, when directly applying the MoCo/PiCO mechanism to multivariate time series, the two encoders (CTFE-W and CTFE-S) receive augmented views whose distributions are much harder to keep aligned than in images, because standard MTS augmentations inevitably distort temporal scales and cross-channel correlations. This leads to severe distribution mismatch between W and S, causing instability, poor convergence, and degraded memory-bank features.
>  To address this, CTFE is not an ad-hoc addition but a principled stabilization mechanism: it introduces a frozen, augmentation-invariant MiniROCKET pathway that is shared by both W and S. This pathway provides a fixed semantic anchor and enforces a consistent latent space even when the augmentations for W and S drift. As a result, the loss landscape becomes smoother and the momentum update remains well behaved, enabling the PiCO-style prototype learning to converge under MTSC-PLL. Ablations show that removing the shared MiniROCKET encoder reduces the model to a MoCo-like structure and leads to a substantial performance drop, confirming that this shared, augmentation-invariant pathway is critical for stable and effective learning in our setting.
>
>
> Weakness 2,3:
>
> Answer:
> PLL setting is synthetic, as partial labels are formed by uniform random thresholding over classes. This is convenient but unlikely to match real ambiguity.
> Many compared methods are not designed for PLL and run with standard settings, while CO-COME explicitly targets PLL. Need adapted PLL variants.
> Answer:
> Our PLL setting is synthetic because, to the best of our knowledge, there is currently no public multivariate time-series benchmark with native partial labels. Following common practice in PLL, we generate candidate label sets by controlled random procedures so that (i) the true label is always included and (ii) the expected candidate-set size can be systematically varied via the partial rate. Importantly, our method also matches or outperforms strong general MTSC models at partial-rate = 0 (see table 1.), and the performance gap widens as the partial rate increases, indicating that the gains are not an artifact of a particular synthetic ambiguity pattern.
> For baselines, we intentionally keep their architectures and objectives close to the original papers and train them under the same candidate-label supervision, instead of engineering separate PLL variants for each model. Constructing “adapted” PLL versions would require additional, method-specific heuristics (e.g., how to aggregate or reweight candidate labels) that are not prescribed by the original works, and their performance would then largely reflect our redesign choices rather than the methods themselves.

---

### Official Review · Reviewer_fQ7r · 2025-11-01

**Soundness:** 2
**Presentation:** 2
**Contribution:** 2
**Rating:** 2
**Confidence:** 3

**Summary:**

This paper presents CO-COME, a method for Multivariate Time Series Classification (MTSC) when labels are ambiguous or noisy.
It uses Partial Label Learning (PLL), where each sample is associated with a set of candidate labels and combines this with contrastive learning and a custom feature encoder.

**Strengths:**

- The problem of inaccurate labels in multivariate time series is an important and interesting problem setting
- I believe applying partial label learning to multivariate time series classification is novel
- The architecture diagrams are well done
- The paper is mostly well written, and usually easy to follow (though there are some egregious formatting errors --- see weaknesses)

**Weaknesses:**

- The formatting of the paper is off. In-text citations often do not have a whitespace between the text of the citation and the character preceding it. Many parenthesis are also missing a whitespace.  Some text appears to have formatting errors caused from being copy and pasted from another source, such as in line 114: "contrastive representa- tion space". These formatting issues do not make the paper unreadable, but they show a lack of polish.

- The method is rather ad hoc; there is no analysis showing that the proposed method will converge to the correct solution. Design choices are not given sufficiently grounded justification

- I may be wrong, but I believe that most of the compared methods are general MVTSC methods, and aren't designed to handle noisy/ambiguous labels. In that case, it's not surprising that the proposed approach outperforms them when there is significant label noise. I believe the authors should compare against time series methods designed to handle noisy labels [1, 2, 3], or at least explain why comparison with them is not reasonable/feasible.

- Discussion of related work focus primarily on PLL methods. More discussion should be given to how ambiguous / missing / noisy labels is handled in time series data, in general.

[1] Nagaraj, Sujay, et al. "Learning under Temporal Label Noise." ICLR 2025.
[2] Zhou, Zhi, Yi-Xuan Jin, and Yu-Feng Li. "Rts: learning robustly from time series data with noisy label." Frontiers of Computer Science 2024.
[3] Atkinson, Gentry, and Vangelis Metsis. "Identifying label noise in time-series datasets." Adjunct proceedings of the 2020 ACM international joint conference on pervasive and ubiquitous computing and proceedings of the 2020 ACM international symposium on wearable computers. 2020.
[4] Cui, Beilei, et al. "Rectifying noisy labels with sequential prior: Multi-scale temporal feature affinity learning for robust video segmentation." International Conference on Medical Image Computing and Computer-Assisted Intervention. Cham: Springer Nature Switzerland, 2023.

**Questions:**

- Which compared methods were designed to handle ambiguous / noisy / missing labels?

---

> ### Author Response · Authors · 2025-11-24
> **Response to weakness 1&2**
>
> Weakness1:
> paper formatting
> A1:
> We sincerely apologize for the formatting issues you mentioned in the paper; we have already corrected them.
> Thank you for the careful review of our manuscript. We apologize for any inconvenience caused by the formatting issues, and all the problems you pointed out have been revised accordingly.
>
> Weakness2:
> The method is rather ad hoc; there is no analysis showing that the proposed method will converge to the correct solution. Design choices are not given sufficiently grounded justification
> A2:
> We respectfully disagree with the comment that our method is “ad hoc”. The overall architecture—especially the integration of CTFE with the PiCO-style contrastive and prototype-based learning—is deliberately designed to address a concrete limitation that arises when applying momentum-based contrastive learning (MoCo/PiCO) to MTSC under partial-label supervision.
>
> First, our method inherits the momentum update mechanism used in MoCo and PiCO, and we explicitly cite both works. Accordingly, we maintain two encoders (CTFE-W and CTFE-S, denoted as W and S networks). However, directly applying this mechanism to MTSC introduces a non-trivial challenge:
> the distributions of augmented views in multivariate time series are much harder to keep consistent than in images, making W and S receive mismatched input distributions. Such distribution mismatch can lead to training instability, poor convergence, and degradation of the memory bank features. This effect is mild in images but becomes substantially amplified in MTSC, where augmentations inherently distort temporal scales or cross-channel correlations.
>
> This motivates our design: we need the model to remain robust when W and S receive heterogeneous augmentations, especially during early training when representations are unstable. At the same time, we must prevent the momentum encoder from collapsing toward the main encoder while ensuring that the loss remains optimizable and convergent.
>
> To achieve this, CTFE is not an ad-hoc addition but a principled solution: it introduces a stable, augmentation-invariant feature pathway shared by both W and S encoders. This pathway does not change during training and serves as a fixed semantic anchor, ensuring that both networks extract features in a consistent latent space regardless of augmentation drift. As a result, the loss landscape becomes smoother, the momentum update remains well-behaved, and prototype-based label disambiguation can converge reliably. Our ablation studies further confirm the necessity of this fixed shared feature extractor. When we remove the MiniRocket-based shared encoder, the model collapses into a structure essentially equivalent to the original MoCo framework, and in this case we observe a substantial performance degradation. This verifies that the shared, augmentation-invariant feature pathway is crucial for stabilizing training and ensuring effective learning under MTSC-PLL conditions.
>
> Regarding the theoretical justification of model convergence, the contrastive learning component follows the PiCO framework, where a detailed convergence analysis has already been provided; we therefore do not repeat it here. Instead, we additionally include a t-SNE visualization of the learned representations colored by labels in Figure 5 of Appendix A.2.4, which we hope helps illustrate the convergence behaviour of our method.

---

> ### Author Response · Authors · 2025-11-24
> **Response to weakness 3**
>
> We thank the reviewer for pointing out the relation to work on learning with noisy labels. However, our problem setting is fundamentally different from the noisy-label (NL) setting considered in [1, 2, 3], and the algorithmic mechanisms used in those works do not transfer to our partial-label (PL) scenario in a principled way.
>
> In the standard NL setting, each training sample is associated with a single (possibly corrupted) label, and the goal is to recover the real true label. In contrast, our task is partial-label MTSC: each X is associated with a set of candidate labels where the ground-truth label is contained in this set. This difference in label formulation is fundamental. NL methods are explicitly designed for the case where the observed label may be wrong and typically introduce mechanisms that correct or reweight these wrong labels (e.g., estimating a label transition matrix, modeling instance-dependent noise, or using special small-loss / confidence-based selection). Partial-label learning instead assumes that all labels outside the candidate set are impossible, and that disambiguation must happen within the candidate set.
>
> Due to the different problem-solving scenarios, the Partial-label learning we proposed cannot be directly compared with NL methods. Furthermore, if we forcibly compare these two types of algorithms by collapsing each candidate label set into a single "proxy label" through additional heuristics or pseudo-labeling, this would violate the fundamental assumptions of partial-label learning.
>
> Moreover，regarding the algorithms mentioned in the reference comparison literature suggested by the reviewers, their assumptions and the problems they address are also inconsistent with those of the Partial-label Learning algorithm we designed.
> For example, [1] considers temporal label noise, where each time step in the sequence has its own label, and noise is modeled as misalalignment of this per-timestep label sequence.
> Our setting is sequence-level classification with a single label (or a candidate label set) per time series, and we do not have time-step-level supervision.
>
> Due to the difference in the data structures of the problems being addressed, adapting [1] would require us to synthesize an artificial label sequence from a single partial label, which is not well-defined and would introduce additional assumptions unrelated to our task.
>
> Similarly, [2],[3] is specifically designed for univariate time-series classification, again with a single observed label per series. Extending [2],[3] to our multivariate time-series setting would require nontrivial architectural and modeling changes on top of the already different label semantics.
> Crucially, once such substantial changes are made, we can no longer guarantee that the modified NL methods preserve either the original accuracy reported by their authors or the methodological intent of those works. The resulting “adapted” baselines would effectively be new algorithms, and any comparison would reflect our particular reimplementation choices rather than the original NL methods themselves, leading to a distorted and potentially misleading comparison.
>
> We again thank the reviewer for pointing us to these excellent works, and we fully acknowledge that [1–3], as well as other noisy-label time-series methods, make valuable contributions within the broader landscape of label ambiguity. In the revision, we have updated the Related Work section to explicitly discuss these methods, add citations to [1–3], and acknowledge their contributions. At a high level, label ambiguity in supervised learning can be roughly divided into three relatively independent directions: (i) missing labels (incomplete supervision), (ii) noisy labels, and (iii) partial labels. Each of these branches comes with its own modeling assumptions and task-specific design choices, and methods are typically equipped with specialized modules and architectures tailored to that branch. Consequently, these methods are often only directly applicable within their own setting, and constructing cross-branch comparisons that are faithful and fair is intrinsically difficult.
>
> We fully understand the reviewer’s concern regarding experimental completeness and fairness. However, systematically bridging these three branches—e.g., by establishing a unified comparison protocol or theoretically justified reductions that map missing-label, NL, and PLL problems into a common framework—remains, to the best of our knowledge, an open and nontrivial research direction. Designing such a unifying standard or equivalence transformation is itself an interesting problem that goes beyond the scope of the present paper.
>
> Let us reiterate the scope and contribution of our work: our method is specifically designed for this PLL–MTSC scenario and, in doing so, fills a gap in the literature and provides an innovative step toward handling label ambiguity in multivariate time-series data.

---

> > ### Comment · Reviewer_fQ7r · 2025-11-27
> >
> > Thank you for your response.
> >
> > I still am not fully convinced that the design choices are sufficiently justified; e.g. there is not sufficient theoretical analysis showing that the augmentation-invariant feature pathway is an optimal / principled solution to the distribution mismatch problem. And I am not convinced that the addition of this pathway is a significant contribution.
> >
> > However, I think my issues regarding prior works has been sufficiently addressed.
> >
> > Due to this, I will raise my score to a 4. I don't think this is a bad paper, and will not fight against its acceptance. But as it stands I do not believe it reaches the technical depth necessary for me to give it a strong endorsement.

---

### Author Response · Authors · 2025-12-04
**Discussion Summary**

The reviewers mainly raise concerns about

(i) formatting issues.

(ii) doubt the MiniROCKET pathway within CO-COME. The reviewer questioning both its impact on convergence behavior and the rationale behind the design choices.

(iii) the fairness and completeness of the experimental comparison, including missing time-series noisy-label baselines and the use of a synthetic PLL setting.

(iv) the role of MiniROCKET, including why we did not evaluate on univariate UCR datasets, whether Table 1 is a fair comparison to MiniROCKET, and whether PiCO/other PLL methods should also be equipped with MiniROCKET.


In response:
1. We have corrected all formatting problems.

2. We integrate the MoCo/PiCO modules as the dual-encoder momentum framework of our model, which exerts a positive effect on information extractor for time-series data and label disambiguation. However, ordinary time series data augmentation simultaneously induces a significant distribution shift between the weak view and strong view. To leverage the advantages brought by the MoCo/PiCO modules while addressing the distribution drift issue, we introduce a shared frozen MiniROCKET pathway, treating it as a stable augmentation-invariant anchor to ensure that CTFE-W (weak-view encoder) and CTFE-S (strong-view encoder) reside in a consistent latent space. To verify the genuine utility of this module, we conduct ablation experiments that demonstrate: removing this shared pathway triggers a MoCo-like model collapse, leading to a distinct performance degradation (particularly pronounced under high partial label rate scenarios); additionally, we supplement representation visualization results to corroborate the convergence characteristics of the model.

3. On experimental fairness, we explain that our problem is partial-label MTSC—fundamentally different in label semantics and supervision granularity from noisy-label time-series settings. So directly adapting NL methods would either violate PLL assumptions or amount to designing new algorithms; instead, we compare against strong, widely used MTSC models under both clean and partial labels, and show that CO-COME is competitive even at partial rate 0, with increasing gains as ambiguity grows. For the synthetic PLL setting, we follow standard PLL practice (true label always in the candidate set, controllable partial rate) due to the lack of native MTSC-PLL benchmarks, and we have updated the Related Work to clearly position noisy-label, missing-label, and partial-label approaches and to cite the suggested papers.

4. Regarding MiniROCKET, we clarifie that we use its multivariate extension as a deterministic shared backbone inside CTFE for MTSC-PLL, not to improve univariate UCR benchmarks, so adding UCR experiments would shift the paper away from its main goal; Table 1 is explicitly a fully supervised MTSC study showing that CO-COME outperforms MiniROCKET and an ablation corresponding to PiCO + MiniROCKET (our CTFE without the CI-encoder) is already included and is consistently weaker than the full CO-COME model under high ambiguity.

Reviewer fQ7r acknowledged that their concerns about prior work and baselines are sufficiently addressed, raised their score to 4, and stated that they will not argue against acceptance. As for reviewers Q2&3, they may not have fully understood some of our work and motivations. There were misunderstandings regarding the experimental setup and results, which we have responded in detail.

For details, you can view the full discussion content. This summary is intended to provide a brief overview of the review and discussion.

Hope this summary may help you.

---

### Meta-Review · Area_Chair_tJqL · 2025-12-24

**Summary:**

This paper introduces CO-COME, a new framework for partial-label multivariate time-series classification. The approach couples contrastive-based label disambiguation with a compact encoder for multivariate temporal representations, allowing the model to effectively handle ambiguous supervision while learning discriminative features. Experimental results on 20 datasets from the UEA archive show that the proposed method consistently outperforms existing approaches in partial-label settings.

**Reviewer Concerns:**

Most reviewers expressed concerns regarding the MiniROCKET pathway in CO-COME, particularly its role and necessity within the partial-label learning framework. Additional issues were raised about the fairness and completeness of the experimental evaluation, including the need for a broader and more appropriate set of partial-label learning baselines and evaluation metrics. Reviewers fQ7r, sh5b, and U7Y3 further noted that the methodological novelty appears incremental, as the overall design largely follows that of PiCO. Moreover, the motivation would be strengthened by validation on real-world multivariate time-series datasets, as reviewer sh5b pointed out that the current partial-label setting is primarily synthetic.

**Reviewer Scores:**

The initial overall scores from reviewers fQ7r, sh5b, WEWm, and U7Y3 were 2, 6, 6, and 2, with corresponding confidence scores of 3, 3, 4, and 4. After considering the full rebuttal and the authors’ consolidated responses, reviewer fQ7r increased the overall score from 2 to 4, while noting that the contribution remains limited. The other reviewers did not participate in further discussion; however, I expect that reviewer U7Y3 may raise the overall score from 2 to 4, whereas the remaining two reviewers are likely to maintain their original evaluations.

Overall, the primary remaining issues concern the lack of clear real-world motivation for partial-label learning and the need to more explicitly articulate the core innovations relative to existing partial-label learning approaches, such as PiCO.

---

### Decision · Program_Chairs · 2026-01-26

Reject